# The Role of Age at Onset on the Clinical Course and Biochemical Parameters of Anorexia Nervosa

**DOI:** 10.3390/jpm15090442

**Published:** 2025-09-17

**Authors:** Lorenzo Ferrario, Andrea Costantino, Letizia Maria Affaticati, Massimo Clerici, Antonios Dakanalis, Enrico Capuzzi, Massimiliano Buoli

**Affiliations:** 1Unit of Gastroenterology and Epatology, Fondazione IRCCS Ca’ Granda Ospedale Maggiore Policlinico, 20122 Milan, Italy; 2Unit of Gastroenterology and Endoscopy, Fondazione IRCCS Ca’ Granda Ospedale Maggiore Policlinico, 20122 Milan, Italy; andrea.costantino@policlinico.mi.it; 3Department of Pathophysiology and Transplantation, University of Milan, 20122 Milan, Italy; massimiliano.buoli@unimi.it; 4Department of Medicine and Surgery, University of Milan Bicocca, via Cadore 38, 20900 Monza, Italy; letizia.affaticati@gmail.com (L.M.A.); massimo.clerici@unimib.it (M.C.); antonios.dakanalis@unimib.it (A.D.); e.capuzzi1@campus.unimib.it (E.C.); 5Department of Mental Health, Fondazione IRCCS San Gerardo dei Tintori, via G.B. Pergolesi 33, 20900 Monza, Italy; 6Department of Neurosciences and Mental Health, Fondazione IRCCS Ca’Granda Ospedale Maggiore Policlinico, 20122 Milan, Italy

**Keywords:** anorexia nervosa, clinical variables, psychiatric comorbidity, cholesterol, Na^+^/K^+^, childhood/adolescence vs. adulthood onset, mortality

## Abstract

**Background:** Anorexia nervosa (AN) has the highest mortality rate among psychiatric disorders, making early diagnosis and tailored management crucial. This study aimed to evaluate the impact of age at onset (AAO)—childhood/adolescence versus adulthood—on the clinical course and associated biochemical parameters. **Methods:** Seventy-six female patients with AN were divided into two groups based on AAO (<18 years vs. ≥18 years). Group comparisons were performed using *t*-tests for continuous variables and χ^2^ tests for categorical variables. Correlation analyses assessed associations between AAO and continuous variables. Significant findings were entered into regression models, including a binary logistic regression with AAO as the dependent variable and a linear regression with significant correlations as predictors. **Results:** The early-onset group showed significantly higher potassium levels and a lower sodium/potassium ratio (Na^+^/K^+^) compared with the late-onset group (potassium: t = 0.93, *p* < 0.01; Na^+^/K^+^: t = 3.39, *p* < 0.01). AAO was strongly inversely correlated with potassium levels (r = −0.75, *p* < 0.01) and positively correlated with cholesterol (r = 0.574, *p* < 0.01) and Na^+^/K^+^ (r = 0.78, *p* = 0.01). Binary logistic regression correctly classified 87% of cases, showing that lower Na^+^/K^+^ was associated with earlier onset (OR = 2.23, *p* = 0.03). Linear regression confirmed significant associations of AAO with cholesterol levels (B = 0.07, *p* = 0.02) and Na^+^/K^+^ (B = 1.68, *p* < 0.01). **Conclusions:** AAO in AN is strongly associated with specific biochemical parameters. Early-onset patients exhibit more severe electrolyte imbalances, while late-onset cases show higher cholesterol levels, suggesting increased cardiovascular risk. These findings emphasize the importance of personalized treatment approaches according to AAO, although further studies are warranted to confirm these results.

## 1. Introduction

Anorexia nervosa (AN) is a complex psychiatric disorder marked by intense fear of weight gain, distorted body image, and restrictive behaviors aimed at weight loss [1]. It is linked to poor quality of life [2] and a mortality rate at least five times higher than that of healthy individuals [3]. AN is now recognized as a complex and multifactorial disorder influenced by the interplay of biological, psychological, and sociocultural factors, which together determine the onset, severity, and progression of the illness [4].

Biological contributions include genetic predisposition, neuroendocrine dysregulation, and alterations in brain circuits involved in reward processing, interoception, and cognitive control. Emerging evidence also points to neuroinflammatory processes, altered hypothalamic–pituitary–adrenal axis functioning, and the dysregulation of gut–brain signalling as potential contributors to the pathophysiology of AN. Psychological and sociocultural factors encompass anxiety, perfectionism, and pressures to conform to idealized body standards. Nutritional and metabolic alterations are also prominent, including deficiencies in vitamins (B12, folate, D), trace elements (zinc, selenium), and altered lipid profiles, which may reflect both the severity and the chronic course of the disorder [5,6]. Such deficiencies not only impact physical health but may also exacerbate neurocognitive impairments, affective disturbances, and reward-processing abnormalities that perpetuate restrictive behaviors.

Diagnostic criteria have evolved across DSM editions: from DSM-III (1980) [7], which first categorized AN as an eating disorder, to DSM-5 (2013) [8], which removed amenorrhea as a requirement and emphasized body image disturbance and restrictive behaviors. Although the DSM-5 has removed amenorrhea as a diagnostic criterion for AN, the psychosocial impact and potential consequences for bone health continue to be significant, particularly in adolescents. The removal of amenorrhea may influence the diagnostic process in males and pre-pubertal females, who would otherwise not meet the previous DSM-IV criteria [9].

Several clinical variables contribute to a poor prognosis, including anxiety comorbidity, low body mass index (BMI), duration of untreated illness (DUI) [10], and longer illness duration [6]. Among these, AAO has emerged as a particularly influential factor, capable of shaping both the biological trajectory and psychosocial outcomes of the disorder. Early-onset AN may disrupt normal neurodevelopment, leading to more pronounced cognitive, emotional, and physiological alterations, whereas later-onset AN is often associated with a lower BMI at initial assessment [11]. Understanding AAO is, therefore, crucial for predicting illness severity, response to treatment, and long-term outcomes. Current guidelines recommend psychotherapy—especially Cognitive Behavioral Therapy (CBT)—as first-line treatment for adults [12]. Among medications, only olanzapine shows limited efficacy, mainly for weight gain [13]. However, a combination of pharmacological, nutritional, and psychosocial interventions is often necessary, especially for patients with complex or treatment-resistant presentations. Early and targeted intervention is critical given the serious cognitive and physical consequences of AN, which include increased susceptibility to infections, reduced bone density, cardiovascular complications, and potential long-term metabolic dysregulation [14].

Epidemiological data show that AN primarily affects females, with about 1% prevalence among adolescent girls and young women [15]. Incidence in males is lower, possibly due to underdiagnosis [15]. Onset typically occurs between ages 12 and 25, peaking around 14 and 18, though cases are increasingly reported in children as young as 8–9 [16]. Adolescence represents a particularly vulnerable period for body image development and identity formation, and sociocultural pressures—amplified by social media exposure, peer comparison, and cultural ideals—may heighten the risk for disordered eating behaviors during this sensitive developmental window [6].

In this framework, AAO is a key clinical variable in AN, influencing illness severity, response to treatment, cognitive development, and biological trajectory. Earlier onset may lead to greater neurodevelopmental and cognitive alterations, longer illness duration, and more pronounced metabolic disturbances, whereas later onset is generally associated with lower BMI at initial assessment and distinct comorbidity patterns. Identifying clinical and biological markers related to AAO could improve early detection, risk stratification, and the personalization of treatment approaches [11].

In light of these considerations, the purpose of the present research is to investigate the role of AAO on the course of AN. Specifically, we examine how AAO may influence BMI, illness duration, psychiatric comorbidity, nutritional status, and cognitive functioning, with the goal of providing insights for precision medicine interventions. By clarifying the relationship between AAO and these clinical and biological variables, our study seeks to support early, targeted strategies for the management of AN and ultimately improve patient outcomes.

## 2. Materials and Methods

### 2.1. Sample and Study Design

This retrospective, observational, cross-sectional study was conducted on a sample of 76 female patients diagnosed with AN. Participants were recruited either during hospitalization at the inpatient clinic of Fondazione IRCCS Ca’ Granda Ospedale Maggiore Policlinico in Milan or as outpatients during follow-up at the Fondazione IRCCS San Gerardo dei Tintori in Monza, over an 18-month period (July 2021–December 2022).

Inclusion criteria required a DSM-5 diagnosis of AN, female sex, age between 17 and 60 years, and the ability and willingness to provide informed consent. Exclusion criteria included (1) malnutrition due to severe medical complications such as renal failure, advanced liver disease, or severe gastrointestinal disorders; (2) intellectual disability; (3) pregnancy or breastfeeding; (4) intake of vitamin B or D supplements within the previous three months; and (5) presence of severe acute inflammatory diseases or other chronic neurological/medical conditions that could significantly influence biochemical parameters.

The study protocol received approval from the local Ethics Committee (Ospedale San Gerardo di Monza, Approval No. 4060, 20 March 2023), and the research was conducted in accordance with the principles of the Declaration of Helsinki (1975, as revised in 2008), Good Clinical Practice, and the EU General Data Protection Regulation (2016/679).

### 2.2. Assessment

Diagnosis of AN and comorbid psychiatric disorders was confirmed by experienced psychiatrists according to DSM-5 criteria and further supported by the Italian version of the Eating Disorder Examination Interview (EDE-17.0D), administered by trained clinicians.

During the initial psychiatric evaluation, a comprehensive set of sociodemographic and clinical variables was collected. These included age, education, employment, and marital status; illness-related variables such as AAO, illness duration, and DUI; treatment setting; family psychiatric history; psychiatric and medical comorbidities; presence of personality disorders; and current psychopharmacological and psychotherapeutic treatments.

Anthropometric measurements were taken under standardized conditions: body weight was recorded with patients wearing light clothing and no shoes using a calibrated digital scale, and height was measured with a stadiometer. BMI was calculated as weight (kg) divided by height squared (m^2^). DUI was defined as the interval between AN onset and the start of an appropriate treatment, according to international guidelines (family therapy for adolescents, cognitive-behavioral therapy for adults, or olanzapine for weight gain). DUI was defined as the interval between the onset of AN and the initiation of guideline-based treatment, following the definition used in previous studies from our research group [8]. While we acknowledge that different definitions exist in the literature depending on the disorder and study design, we chose this approach for consistency with our prior work.

It should be noted that participants were recruited from specialized clinics, which may represent more severe cases of AN and could limit the generalizability of these findings to community-based or less severe patient populations.

### 2.3. Blood Collection

Venous blood samples were collected under fasting conditions. Inpatients were sampled between 8:00 and 10:00 a.m. at admission, while outpatients were sampled during the morning at their first psychiatric or medical assessment.

Samples were processed following standardized hospital protocols. Haematological parameters, including red and white blood cells, lymphocytes, neutrophils, platelets, haemoglobin, mean corpuscular volume, neutrophil-to-lymphocyte ratio (NLR), and platelet-to-lymphocyte ratio (PLR), were analyzed using an automated haematology analyzer (Sysmex XN-Series, Sysmex Corporation, Kobe, Japan). Biochemical parameters, including sodium, potassium, Na^+^/K^+^ ratio, iron, folate, vitamin D, vitamin B12, and total cholesterol (TC), were assayed on a Roche Cobas 8000 modular analyzer (Roche Diagnostics, Basel, Switzerland) using standardized enzymatic and immunoassay methods. Daily calibration and internal quality controls were performed according to the manufacturer’s instructions.

To ensure reliability, only baseline laboratory results obtained prior to any treatment modification were included, and results were cross-checked with electronic medical records whenever possible.

### 2.4. Statistical Analyses

All statistical analyses were conducted using SPSS software (version 29.0, IBM Corp., Armonk, NY, USA). Descriptive statistics were calculated for all variables, with categorical variables presented as frequencies and percentages, and continuous variables as means and standard deviations.

Participants were divided into two groups based on AAO: <18 years (early-onset) and ≥18 years (late-onset). The groups were compared by unpaired sample *t* tests for continuous variables and χ^2^ tests with odds ratio (OR) and 95% confidence interval (CI) calculation for qualitative variables. The statistically significant variables from *t*-tests were inserted as predictors in a binary logistic regression model with early versus late AAO as the dependent variable.

Correlation analyses were then performed between AAO and continuous variables. Those found to be statistically significant in the correlation analyses were inserted as independent variables in a linear regression model with AAO as the dependent factor.

A *p* ≤ 0.05 was adopted as a statistically significant threshold for all statistical analyses.

A multiple imputation method was used to estimate values for variables with a large number of missing values (sodium, potassium, Na^+^/K^+^ ratio, red blood cells, mean corpuscular volume, haemoglobin, NLR, neutrophils, iron, vitamin D, folate, vitamin B12, cholesterol, type of current medical comorbidity). In addition, post-hoc power analyses were conducted for key outcomes, revealing that analyses related to cholesterol were underpowered (17%, α = 0.05), whereas the analysis of the Na^+^/K^+^ ratio reached good power (90.6%).

## 3. Results

The study sample consisted of 76 female patients affected by AN. Among these, 12 (15.8%) were hospitalized in the inpatient clinic for intensive residential treatment, while the remaining 64 (84.2%) received outpatient care with periodic visits and continuous monitoring, reflecting a wide spectrum of illness severity and management approaches.

The mean age of the participants was 23.19 years (±8.32), with an age range from 18 to 40 years. The mean AAO of the disorder was 18.42 years (±6.48), indicating a prevalence of the disorder at a young age, with onset coinciding with life transition to adolescence or early adulthood. The mean BMI was 17.21 kg/m^2^ (±2.21), significantly lower than the normal range, reflecting the severity of the eating disorder.

Thirteen subjects were in treatment with pharmacological compounds: 1 with sertraline, 2 with escitalopram, 1 with olanzapine, 2 with quetiapine, 1 with aripiprazole, 1 with mirtazapine, 2 with asenapine, 2 with duloxetine, and 1 with haloperidol. Only 6 out of 76 female patients were treated with medications potentially influencing metabolism (mirtazapine, olanzapine, quetiapine, and asenapine).

Table 1 and Table 2 summarize the descriptive analyses of the total sample and of the groups identified by AAO, as well as the results of the *t*-tests and χ^2^ tests, respectively. Early versus late-onset patients resulted as follows: to be younger (t = 2.86, *p* < 0.01), to have less years of education (t = 3.10, *p* = 0.01), to have a longer duration of illness at borderline statistical significance (t = 1.88, *p* = 0.06), to have higher K+ levels (t = 3.63, *p* < 0.01) and consequently lower Na^+^/K^+^ (t = 3.39, *p* < 0.01), to be more frequently students (χ^2^ = 12.23, *p* = 0.02), to have less frequently comorbidity with depressive disorders (χ^2^ = 10.84, *p* = 0.03) or medical comorbidity (χ^2^ = 13.63, *p* = 0.02).

The goodness-of-fit test (Hosmer and Lemeshow Test: χ^2^ = 7.62, *p* = 0.47) and omnibus test (χ^2^ = 11.97, *p* < 0.01) showed that the model including duration of illness and Na^+^/K^+^ as possible predictors of early versus late AAO was reliable, allowing for a correct classification of 87.0% of the cases. A higher Na^+^/K^+^ ratio predicted later AAO (*p* = 0.03) (Table 3). When medical comorbidities were included as covariates, the Na^+^/K^+^ ratio continued to be statistically significant (OR = 2.17, *p* = 0.03).

A significant direct correlation was found between AAO and the following: age (Pearson’s r = 0.66, *p* < 0.01), years of education (r = 0.30, *p* < 0.01), Na^+^/K^+^ (r = 0.78, *p* < 0.01), and cholesterol plasma levels (r = 0.57, *p* < 0.01) (Table 4).

The linear regression model resulted in being reliable (Durbin–Watson: 2.25). Cholesterol (*p* = 0.02) and Na^+^/K^+^ (*p* < 0.01) were found to have significant associations with AAO (Table 5, Figure 1 and Figure 2). After applying a multiple imputation method for missing values, the results did not change and continued to be statistically significant (*p* < 0.05).

## 4. Discussion

The results of this study indicate that the AAO in AN is significantly associated with several clinical and biochemical parameters. Overall, our findings demonstrate that earlier versus later AAO influences multiple aspects of health and psychosocial functioning, including electrolyte balance, cholesterol plasma levels, educational attainment, employment status, and psychiatric and medical comorbidities, highlighting the multidimensional impact of onset timing on both biological and psychosocial outcomes. In particular, we observed a significant relationship between AAO and Na^+^/K^+^ ratio, cholesterol levels, years of education, employment status, and the presence of psychiatric and medical comorbidities.

The association between an earlier AAO and a lower Na^+^/K^+^ may indicate that patients with childhood or adolescent onset experience more pronounced electrolyte imbalances, reflecting a more severe metabolic dysregulation. This could potentially involve alterations in the renin–angiotensin–aldosterone system, which were documented in women with AN, particularly when medical supervision is limited or delayed [17,18].

Furthermore, even though there is no robust evidence of this phenomenon in the scientific literature, it is known that some individuals with AN tend to misuse dietary supplements containing electrolytes such as K^+^ [19]. It is, therefore, plausible to hypothesize that the observed electrolyte imbalances may, at least in part, be attributed to a pathological compensatory mechanism related to the disorder. This hypothesis is further supported by the fact that many individuals with AN engage in extreme exercise regimens, often combined with excessive use of dietary supplements, in an effort to enhance physical performance and, indirectly, increase energy expenditure [20]. Moreover, as mentioned above, abnormalities in Na^+^ and K^+^ levels serve as indirect indicators of alterations in the renin–angiotensin–aldosterone system, a condition well documented in women with AN and potentially more pronounced in those who remain without adequate medical supervision for extended periods [18].

It should be emphasized that these proposed mechanisms are speculative, and direct measurements of the renin–angiotensin–aldosterone system or biochemical markers were not performed; these hypotheses are supported by relevant literature and should be interpreted cautiously with respect to clinical significance, including potential effects on cardiac rhythm and mental status [21,22].

Blood cholesterol levels also appear to be influenced by AAO, suggesting that patients with late-onset AN may be at greater risk of developing metabolic-related cardiovascular complications over the course of the illness [23]. This finding may be explained by the fact that during adolescence and early adulthood, women undergo significant hormonal changes that influence lipid metabolism, with oestrogens being protective against dyslipidemia [24]. Variations in estrogen production and thyroid function associated with AN may therefore have a greater impact on women with later AAO, potentially affecting both lipid metabolism and overall metabolic health [25]. It is important to note that although elevated cholesterol levels were observed in late-onset AN patients, the link with long-term cardiovascular risk remains complex and may be influenced by confounding factors such as malnutrition, hormonal alterations, and body composition. Therefore, transient biochemical changes should not be interpreted as directly related to increased clinical risk, and caution is warranted when interpreting these results. Of note, cholesterol plays important roles in mental health, contributing to the integrity of the central nervous system [26] and being a precursor of important molecules such as vitamin D, which regulates dopamine pathways [27]. It should be noted that in our study, only TC was measured, without evaluating a complete lipid profile including high-density lipoprotein (HDL), low-density lipoprotein (LDL), and triglycerides. This restricts the assessment of cardiovascular risk and overall metabolic status. Future studies should include a comprehensive lipid panel to better characterize metabolism in patients with AN. Moreover, recent literature suggests that lipid profiles in AN can vary considerably depending on illness severity, nutritional status, and AAO, highlighting the importance of a more detailed analysis [28,29].

It should be noted that in our sample, women with late AAO presented more psychiatric comorbidity, particularly depressive disorders. In this sub-group of patients, the interaction between metabolic alterations such as elevated cholesterol levels, age-related hormonal fluctuations, and the psychosocial challenges of adulthood may increase susceptibility to mood disturbances. Additionally, the presence of medical comorbidities and the overlapping symptoms between depression and AN—such as reduced food intake, sleep disturbances, and cognitive impairments—can further exacerbate psychological distress, complicate treatment strategies, and contribute to a more severe and prolonged clinical course. Taken as a whole, these factors highlight the need for a comprehensive, multidisciplinary approach tailored to the specific vulnerabilities of late-onset AN patients. Of note, psychiatric comorbidities are conditions that may occur before or after the onset of AN, increasing the severity of the illness and potentially complicating treatment management and future prognosis [30]. As mentioned above, the frequent co-occurrence of depression in patients with late-onset AN makes their management challenging due to the overlap of symptoms, such as reduced food intake or insomnia [31]. These issues are further exacerbated by the fact that patients with late-onset AN are more likely to present medical comorbidities, which, in turn, increase the vulnerability to psychiatric disorders and contribute to a more severe and complex illness trajectory [32]. This supports the necessity for careful screening and early intervention for psychiatric and medical comorbidities, particularly in patients with late-onset AN, to prevent further deterioration and improve long-term outcomes.

Finally, the differences in years of education between the two AAO groups could simply be explained by the fact that many patients are still in school. However, they suggest that early-onset (childhood/adolescence) AN may be associated with developmental and relational issues, thereby influencing the course and management of the disorder. The education level of patients with AN can be a critical aspect to examine in relation to the course of the illness. Of note, a recent article examined the educational level of patients’ parents, finding that subjects with parents with high educational levels had a longer length of stay during hospitalizations, less dietary restraint, and higher personal standards [33].

Taken together, these results highlight that AAO is a critical determinant of clinical, metabolic, and psychosocial profiles in AN. Early-onset AN would be associated with more pronounced electrolyte dysregulation and developmental challenges, whereas late-onset AN would be linked to higher cholesterol plasma levels and increased depression comorbidity. Recognizing these patterns enables clinicians to implement more individualized treatment strategies, including targeted nutritional, pharmacological, and psychotherapeutic interventions.

In conclusion, AAO should be considered a central factor in the evaluation and management of AN, with implications for biological monitoring, psychosocial support, and the personalization of treatment. Future research should further investigate the mechanisms linking AAO to metabolic, hormonal, and psychological outcomes to refine precision medicine approaches in eating disorders.

## 5. Study Limitations

The results of this study must be interpreted with caution, considering several methodological limitations. A first limitation is the retrospective nature of data collection, which in some cases made it difficult to retrieve detailed information on specific clinical aspects of the patients, although computerized medical records and regional databases facilitated this task.

Another major issue is the amount of missing data for some variables. To address this aspect, we quantified the missing values using a multiple imputation method. A further limitation concerns the statistical power of the analyses. Post-hoc calculations indicated that the power for cholesterol-related analyses was only 17% (α = 0.05), confirming that the sample was underpowered for this outcome. In contrast, the analysis of the Na^+^/K^+^ ratio reached a power of 90.6%, suggesting that findings in this domain are more robust. Nevertheless, the small sample size and missing data increase the risk of Type II error and may lead to spurious associations, thus warranting caution in the interpretation of the results.

In addition, some variables were not routinely collected in one of the participating centers, further reducing completeness. The relatively small sample size may also have limited statistical power and generalizability. It should be noted that the findings are based primarily on patients from specialized clinics, who may represent more severe cases, thus limiting generalizability to community-based patients or those with less severe forms of AN. Future studies should include more heterogeneous samples from different settings to confirm these results. Moreover, the clinical characteristics and illness severity of our sample may not be fully representative of patients with AN who do not access specialized services or those with milder forms of the disorder. Furthermore, the cross-sectional and retrospective design may have introduced selection or information bias, and potential recall bias for self-reported data (e.g., AAO) could have influenced the results.

Some clinical data were collected during the COVID-19 pandemic, a context that may have negatively affected the course of eating disorders [34,35]. In our study, only TC was measured, without including a complete lipid profile, and only total vitamin B12 was evaluated, without assessing other biomarkers useful for a more in-depth analysis of cobalamin status, such as holotranscobalamin [36] or other markers [37]. Future work is planned to expand the analysis by measuring additional inflammatory and metabolic markers, including Tumor Necrosis Factor-α, IL-6, C-reactive protein, and other antioxidants, to provide a more comprehensive biochemical characterization.

Another limitation concerns the diagnosis of AN severity, which was based solely on BMI, in accordance with the DSM-5. However, severity is a more complex concept, encompassing not only the rate of weight loss, but also body composition and psychological aspects such as body image distortion [38]. Some subjects were undergoing pharmacological treatment. While the effectiveness of these drugs in weight recovery remains debated [39], they may positively impact symptoms such as anxiety, compulsive behaviors, and depression [40]. Nevertheless, pharmacotherapy or substance use disorders may significantly affect weight and metabolism, potentially influencing biochemical parameters such as total cholesterol, and thus representing a confounding factor.

Lastly, despite excluding patients taking vitamin B or vitamin D, the use of other supplements or substances may have altered some biomarker values, representing a further confounder. Taken together, these limitations—including retrospective design, missing data, small sample size, limited biochemical assessment, and potential pharmacological influences—may have reduced the reliability of our results. Therefore, caution is warranted when interpreting these findings and generalizing them to broader AN populations. Future longitudinal studies on larger, more diverse samples, including both anthropometric and psychological parameters, are needed to identify predictive factors of AN severity and to optimize preventive and therapeutic strategies.

## 6. Conclusions

Our analysis highlighted how patients, depending on their AAO, may have different clinical and metabolic profiles that require specific treatment (electrolyte imbalance for earlier AAO and dyslipidemia for later AAO individuals). Considering the clinical implications arising from our research, it becomes clear that an integrated and multidisciplinary therapeutic approach is essential to effectively address the management of AN. As well as a pharmacological or psychotherapeutic approach, according to patients’ AAO, it will be necessary to take into consideration the prescription of targeted diets to prevent electrolyte imbalance and hypercholesterolemia, or complementary treatments such as yoga to address psychiatric/medical comorbidity [27,41].

Future research will have to confirm the results of the present article, and studies with larger samples are necessary to clarify the role of AAO in the outcome of patients affected by AN.

These results highlight the importance of tailoring therapeutic approaches according to the AAO of AN to optimize interventions and improve outcomes for different patient groups. Our findings showed that patients with early-onset and late-onset AN present distinct clinical and metabolic profiles, requiring different management strategies. For example, early-onset patients may need more focused interventions to correct electrolyte imbalances, while late-onset patients could benefit from strategies addressing dyslipidemia and cardiovascular risk [42]. Understanding these differences can help clinicians to anticipate potential complications and adapt monitoring and prevention plans accordingly.

Considering the clinical implications of our results, it is clear that an integrated, multidisciplinary approach is essential for effective management of AN. In addition to standard pharmacological or psychotherapeutic treatments tailored to AAO, clinicians should consider nutritional strategies, such as customized diets to prevent electrolyte disturbances, correct micronutrient deficiencies, and manage high cholesterol. Complementary practices, including yoga, mindfulness, or structured physical activity, could support mental well-being and help address psychiatric and medical comorbidities [27,41]. Implementing such strategies in a coordinated care plan may enhance adherence, improve quality of life, and reduce the likelihood of relapse.

Moreover, our study emphasizes the importance of early identification of AAO-specific risk factors and of continuous monitoring, which may guide preventive and precision medicine approaches. For instance, late-onset AN patients may require closer attention to mood disturbances, cardiovascular health, and medical comorbidities, while early-onset patients may benefit from interventions supporting cognitive development and neurobiological resilience alongside nutritional rehabilitation.

Future research should aim to confirm and expand these findings through longitudinal studies with larger and more heterogeneous populations, including detailed metabolic, hormonal, and psychological assessments. This would provide deeper insight into the mechanisms linking AAO with clinical trajectories and help to refine personalized treatment plans, ultimately improving prognosis and long-term outcomes for patients with AN. Overall, integrating data on AAO into clinical practice offers a practical and promising tool for enhancing individualized care and optimizing the management of this complex disorder.

## Figures and Tables

**Figure 1 jpm-15-00442-f001:**
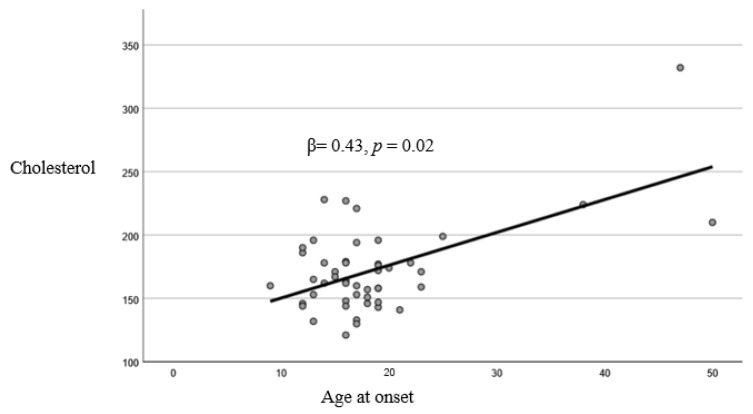
Linear relationship between cholesterol and age at onset.

**Figure 2 jpm-15-00442-f002:**
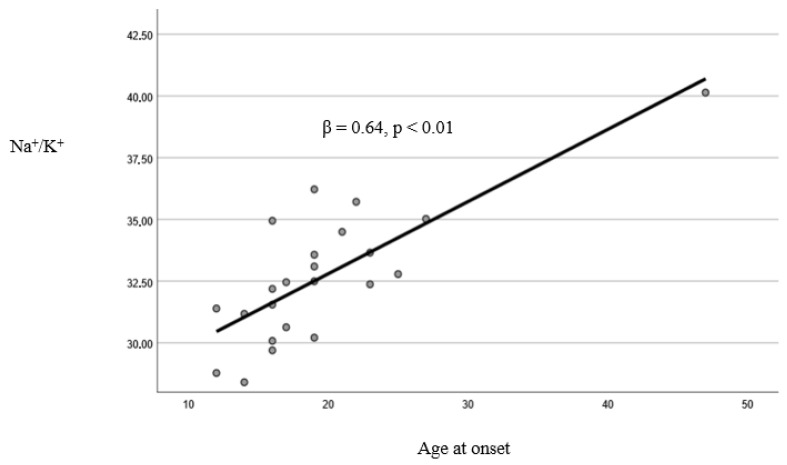
Linear relationship between the sodium/potassium (Na^+^/K^+^) ratio and age at onset.

**Table 1 jpm-15-00442-t001:** Continuous clinical variables and biochemical parameters of the total sample and of the two groups identified by age at onset.

Variables	Total Sample *n* = 76	Early Onset (<18 Years) *n* = 42	Late Onset (≥18 Years) *n* = 34	t	*p*
**Age** **Missing *n* = 0**	23.12 ± 8.32	20.69 ± 6.48	26.12 ± 9.40	2.86	**<0.01**
**DUI (years)** **Missing *n* = 0**	2.11 ± 2.71	2.26 ± 2.27	1.92 ± 3.15	0.54	0.59
**Duration of illness (years)** **Missing *n* = 0**	4.17 ± 5.81	5.28 ± 7.01	2.80 ± 3.45	1.88	0.06
**Educational status (years)** **Missing *n* = 4**	12.21 ± 3.04	11.26 ± 3.05	13.33 ± 2.64	3.10	**0.01**
**BMI** **Missing *n* = 4**	17.23 ± 2.23	17.25 ± 1.92	17.21 ± 2.57	0.07	0.95
**Na^+^ (mmol/L)** **Missing *n* = 52**	140.33 ± 2.78	140.99 ± 2.58	139.56 ± 2.91	1.08	0.29
**K^+^ (mmol/L)** **Missing *n* = 53**	4.33 ± 0.35	4.55 ± 0.28	4.12 ± 0.29	3.63	**<0.01**
**Na^+^/K^+^** **Missing *n* = 53**	32.66 ± 2.68	31.03 ± 1.84	34.15 ± 2.49	3.39	**<0.01**
**Red blood cells (10^6^/mm** **^3^)** **Missing *n* = 50**	4.27 ± 0.44	4.19 ± 0.42	4.32 ± 0.46	0.76	0.45
**MCV (fL)** **Missing *n* = 50**	90.82 ± 5.47	90.39 ± 4.50	91.14 ± 6.21	0.34	0.74
**Hb (g/dL)** **Missing *n* = 50**	13.05 ± 1.26	12.80 ± 1.44	13.23 ± 1.13	0.85	0.41
**Leukocytes (10^9^/L)** **Missing *n* = 2**	5.67 ± 1.76	5.80 ± 1.81	5.51 ± 1.71	0.71	0.48
**Lymphocytes (10^9^/L)** **Missing *n* = 3**	2.13 ± 0.75	2.23 ± 0.81	2.00 ± 0.64	1.28	0.20
**Neutrophils (10^9^/L)** **Missing *n* = 29**	3.23 ± 1.70	3.25 ± 1.77	3.20 ± 1.63	0.11	0.91
**Platelets (10^9^/L)** **Missing *n* = 2**	227.66 ± 49.34	232.53 ± 45.76	221.94 ± 53.37	0.92	0.36
**NLR** **Missing *n* = 29**	1.55 ± 0.94	1.37 ± 0.69	1.83 ± 1.20	1.49	0.15
**PLR** **Missing *n* = 3**	117.06 ± 41.06	114.04 ± 38.26	120.72 ± 44.54	0.69	0.49
**Iron (** **μg/dL)** **Missing *n* = 20**	106.60 ± 74.72	110.73 ± 87.67	101.82 ± 57.64	0.64	0.66
**Vitamin D (ng/mL)** **Missing *n* = 34**	30.58 ± 14.24	29.92 ± 14.41	31.46 ± 14.39	0.34	0.73
**Folate (ng/mL)** **Missing *n* = 21**	8.86 ± 6.24	7.91 ± 3.69	10.00 ± 8.29	1.17	0.25
**Vitamin B12 (pg/mL)** **Missing *n* = 26**	529.42 ± 325.68	553.03 ± 376.15	500.92 ± 256.90	0.58	0.57
**Cholesterol (mg/dL)** **Missing *n* = 28**	172.10 ± 34.94	167.57 ± 28.13	178.45 ± 42.69	1.07	0.29

**Legend**: BMI: body mass index; DUI: duration of untreated illness; Hb: haemoglobin; K+: potassium; MCV: mean corpuscular volume; Na+: sodium; NLR: neutrophil/lymphocyte ratio; PLR: platelet/lymphocyte ratio; *p*: *p* value; t=: Student’s t value. Mean and ± standard deviations are reported. In bold, statistically significant *p* and missing values. Reference values for key biochemical and haematological parameters for adult female participants are provided to allow the reader to compare observed patient values with established physiological norms: sodium (**Na^+^**) 135–145 mmol/L, potassium (**K^+^**) 3.5–5.0 mmol/L, **Na^+^/K^+^ ratio** 27–35 (physiological estimate), red blood cells (**RBC**) 4.0–5.2 × 10^6^/mm^3^, mean corpuscular volume (**MCV**) 80–96 fL, hemoglobin (**Hb**) 12.0–16.0 g/dL, leukocytes (**WBC**) 4.5–11.0 × 10^9^/L, **lymphocytes** 1.0–4.8 × 10^9^/L, **neutrophils** 2.5–7.0 × 10^9^/L, **platelets** 150–400 × 10^9^/L, neutrophil-to-lymphocyte ratio (**NLR**) 0.78–3.53, platelet-to-lymphocyte ratio (**PLR**) not standardized, iron 50–170 µg/dL, **vitamin D** ≥ 30 ng/mL (optimal 30–100), **folate** 5–25 ng/mL, **vitamin B12** 200–800 pg/mL, and **total cholesterol** < 200 mg/dL.

**Table 2 jpm-15-00442-t002:** Qualitative clinical variables of the total sample and of the two groups identified by age at onset.

Variables		Total Sample *n* = 76	Early Onset (<18 Years) *n* = 42	Late Onset (≥18 Years) *n* = 34	χ^2^	OR (95% CI)	*p*
**Current partner** **Missing *n* = 10**	**No**	47 (71.2%)	21 (63.6%)	26 (78.8%)	1.85	0.47 (0.16–1.41)	0.28
	**Yes**	19 (28.8%)	12 (36.4%)	7 (21.2%)			
**Work status** **Missing *n* = 3**	**Unemployed**	13 (17.8%)	10 (25%)	3 (9.1%)	12.23	/ *	**0.02**
	**Student**	49 (67.1%)	29 (72.5%)	20 (60.6%)			
	**Worker**	11 (15.1%)	1 (2.5%)	10 (30.3%)			
**Lifetime substance use disorders** **Missing *n* = 2**	**No**	65 (87.8%)	36 (90.0%)	29 (85.3%)	0.38	1.55 (0.38–6.31)	0.72
	**Yes**	9 (12.2%)	4 (10.0%)	5 (14.7%)			
**Lifetime poly-substance use disorders** **Missing *n* = 3**	**No**	70 (95.9%)	38 (95.0%)	32 (97.0%)	0.51	2.38 (0.21–27.42)	0.59
	**Yes**	3 (4.1%)	2 (5.0%)	1 (3.0%)			
**Psychiatric family history** **Missing *n* = 2**	**No**	49 (66.2%)	25 (62.5%)	24 (70.6%)	0.54	0.69 (0.26–1.84)	0.62
	**Yes**	25 (33.8%)	15 (37.5%)	10 (29.4%)			
**Current psychotherapy** **Missing *n* = 14**	**No**	58 (93.5%)	32 (97%)	26 (89.7%)	1.37	3.69 (0.36–37.63)	0.33
	**Yes**	4 (6.5%)	1 (3.0%)	3 (10.3%)			
**Psychiatric comorbidity** **Missing *n* = 0**	**No**	51 (67.1%)	29 (69.0%)	22 (64.7%)	0.16	1.22 (0.47–3.18)	0.80
	**Yes**	25 (32.9%)	13 (31.0%)	12 (35.3%)			
**Type of psychiatric comorbidity** **Missing *n* = 0**	**None**	51 (67.1%)	29 (69.0%)	22 (64.7%)	10.84	/ *	**0.03**
	**Depression**	10 (13.2%)	2 (4.8%)	8 (23.5%)			
	**Anxiety disorders**	8 (10.5%)	4 (9.5%)	4 (11.8%)			
	**OCD**	2 (2.6%)	2 (4.8%)	0 (0.0%)			
	**Other**	5 (6.6%)	5 (11.5%)	0 (0.0%)			
**Psychiatric poly-comorbidity** **Missing *n* = 0**	**No**	68 (89.5%)	37 (88.1%)	31 (91.2%)	0.19	0.72 (0.16–3.24)	0.73
	**Yes**	8 (10.5%)	5 (11.9%)	3 (8.8%)			
**Personality disorder** **Missing *n* = 4**	**None**	63 (87.4%)	32 (84.3%)	31 (91.2%)	8.82	/ *	0.06
	**Schizoid**	1 (1.4%)	0 (0.0%)	1 (2.9%)			
	**Schizotypal**	2 (2.8%)	0 (0.0%)	2 (5.9%)			
	**Borderline**	4 (5.6%)	4 (10.5%)	0 (0.0%)			
	**OCD**	1 (1.4%)	1 (2.6%)	0 (0.0%)			
	**PD-NOS**	1 (1.4%)	1 (2.6%)	0 (0.0%)			
**Thyroid disorders** **Missing *n* = 10**	**No**	65 (98.5%)	38 (97.4%)	27 (100%)	0.70	0.58 (0.48–0.72)	1.00
	**Yes**	1 (1.5%)	1 (2.6%)	0 (0%)			
**Hypercholesterolemia** **Missing *n* = 7**	**No**	66 (95.7%)	37 (94.9%)	29 (96.7%)	0.31	0.64 (0.06–7.39)	1.00
	**Yes**	3 (4.3%)	2 (5.1%)	1 (3.3%)			
**Diabetes** **Missing *n* = 8**	**No**	68 (100%)	39 (100%)	29 (100%)	/ **	/ **	/ **
	**Yes**	0 (0.0%)	0 (0.0%)	0 (0.0%)			
**Current medical poly-comorbidity** **Missing *n* = 4**	**No**	69 (95.8%)	40 (97.6%)	29 (93.5%)	0.71	2.76 (0.24–31.89)	0.57
	**Yes**	3 (4.2%)	1 (2.4%)	2 (6.5%)			
**Type of current medical comorbidity** **Missing *n* = 27**	**None**	37 (75.6%)	26 (86.7%)	11 (57.9%)	13.63	/ *	**0.02**
	**CRF**	1 (2.0%)	1 (3.3%)	0 (0.0%)			
	**GERD**	2 (4.1%)	0 (0.0%)	2 (10.5%)			
	**Hyperthyroidism**	1 (2.0%)	0 (0.0%)	1 (5.3%)			
	**Thalassemia**	2 (4.1%)	2 (6.7%)	0 (0.0%)			
	**Endocrine disorders**	3 (6.2%)	1 (3.3%)	2 (10.5%)			
	**Blood disorders**	1 (2.0%)	0 (0.0%)	1 (5.3%)			
	**Others**	2 (4.0%)	0 (0.0%)	2 (10.5%)			

**Legend**: CI: confidence interval; χ^2^: chi-square; CRF: Chronic Renal Failure; GERD: Gastroesophageal Reflux Disease; OCD: Obsessive Compulsive Disorder; OR: odds ratio; *p*: *p* value; PD-NOS: Personality Disorder Not Otherwise Specified. Frequencies with percentages in brackets are reported. In bold, statistically significant *p*. * = Risk estimate statistics cannot be calculated. These are only computed for a 2 × 2 table with no empty cells. ** = Not calculated because no subjects in the sample suffer from diabetes.

**Table 3 jpm-15-00442-t003:** Summary of the results from the binary logistic regression model.

Variables	B	SE	*p* Value	OR	95% CI	
					Inferior	Superior
**Duration of illness**	−0.17	0.19	0.36	0.84	0.58	1.21
**Sodium/Potassium**	0.80	0.37	**0.03**	2.23	1.1	4.57

**Model characteristic: R^2^ = 0.41; VIF = 1.28**. **Legend**: B: regression coefficient; CI: confidence interval; OR: odds ratio; SE: standard error. The dependent variable is represented by early versus late age at onset. In bold are statistically significant differences (*p* < 0.05).

**Table 4 jpm-15-00442-t004:** Significant correlations between age at onset and continuous variables.

Variables Correlated with Age at Onset	r	*p*
**Age (years)**	0.66	<0.01
**Education level (years)**	0.30	<0.01
**K^+^ (mmol/L)**	−0.75	<0.01
**Na^+^/K^+^**	0.78	<0.01
**Cholesterol (mg/dL)**	0.57	<0.01

**Legend**: K^+^: potassium; Na^+^: sodium; r: Pearson’s r; *p* = *p* value.

**Table 5 jpm-15-00442-t005:** Linear regression model with age at onset as the dependent variable.

Variable	B	SE	β	*p*
**Cholesterol**	0.07	0.02	0.43	**0.02**
**NA^+^/K^+^**	1.68	0.31	0.64	**<0.01**

**Model characteristic: R^2^ = 0.74; VIF = 1.1**. **Legend**: B: regression coefficient; β: standardized regression coefficient; K+: potassium; NA+: sodium; *p*: *p* value; SE: standard error. Statistically significant *p* values are shown in bold. β = standardized coefficient *p* = *p* value.

## Data Availability

The raw data supporting the conclusions of this article will be made available by the authors upon request.

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
