# Peer review of "The Role of Age at Onset on the Clinical Course and Biochemical Parameters of Anorexia Nervosa"

_jpm, 2025, doi:10.3390/jpm15090442_

Round 1
Reviewer 1 Report
Comments and Suggestions for Authors
This manuscript investigates the impact of age at onset (AAO) on the clinical course and biochemical parameters of anorexia nervosa (AN) among 76 female patients, divided into early onset (<18 years) and late onset (≥18 years) groups. Using a retrospective observational design, the authors assessed demographic, clinical, and biochemical variables at baseline, including electrolyte levels and cholesterol. Statistical analyses involved t-tests, chi-squared tests, correlation analyses, and regression modeling. Key findings show that earlier onset AN is associated with greater electrolyte disturbances (notably lower sodium/potassium ratio and higher potassium), whereas later onset correlates with elevated cholesterol, indicative of increased cardiovascular risk. Additionally, the late onset group presented more comorbid depression and medical conditions. The authors conclude that AAO influences distinct clinical and metabolic profiles, emphasizing the need for personalized treatment approaches based on AAO. The manuscript is interesting and rather novel but includes a number of limitations, such as retrospective data collection, missing values for some biochemical markers, single-center recruitment for certain variables, and the impact of the COVID-19 pandemic on data collection. Below is a list of major and minor issues that needs to be corrected
Major Issues 1. Retrospective Design & Missing Data; Retrospective studies are prone to bias and incomplete data. Significant missing data for certain biochemical markers (e.g., 50 missing values for RBC, MCV, Hb) undermine the statistical power and generalizability. Authors should quantify missing data for each variable and apply imputation methods or sensitivity analyses. 2. Sample Size & Statistical Power; A sample of 76, further subdivided, especially with large numbers of missing data, is underpowered for regression analyses. This limitation should be addressed explicitly, including a post-hoc power calculation. The authors must acknowledge that small sample size and missing data inflate the risk of Type II error and may lead to spurious associations. 3. Generalizability; the focus on specialized clinics restricts the findings to more severe cases, potentially limiting applicability to community-based or less severe patients. This should be discussed more thoroughly, and future studies should be proposed that include samples from diverse settings. 4. Definition and Use of Duration of Untreated Illness (DUI); The definition differs across studies; some consider the initiation of any intervention, not necessarily guideline-based. Authors should clarify their definition, compare it to recent literature, and discuss implications. Further, consider the role of subjective versus objective diagnosis dates. 5. Discussion, Interpretation of Electrolyte Imbalances; The discussion provides plausible but speculative mechanisms (e.g., renin-angiotensin system involvement, supplement misuse) without direct measurement or supporting biochemical markers. Authors should note these are hypotheses and cite relevant literature for each mechanism. Additionally, the clinical significance should be appraised (incident cardiac arrhythmias, mental status changes). and ensure recent publications are considered: https://jamanetwork.com/journals/jamanetworkopen/fullarticle/2798207 6. Discussion, Cholesterol and Cardiovascular Risk; The link between cholesterol and cardiovascular risk in AN is controversial; some reviews caution against extrapolating risk from hypercholesterolemia in AN due to confounders like malnutrition. It is important to clarify the distinction between transient biochemical changes and long-term clinical risk. see also next point (7). 7. Limitations, Incomplete Lipid Profile; "...only total cholesterol (TC) was measured, without including a complete lipid profile..." The lack of HDL, LDL, and triglyceride data limits the evaluation of cardiovascular risk and metabolic status. Authors should recommend complete lipid profiling in future studies and acknowledge this as a major limitation. In addition, they should review and cite recent literature that specifically investigate lipid profiles in AN and include this, also in relation to their interpretation. the authors should consider these publications: https://pmc.ncbi.nlm.nih.gov/articles/PMC6842568/ , https://scielo.isciii.es/pdf/nh/v27n3/24_original21.pdf 8. Introduction, DSM-5 Diagnostic Criteria Adaptation; While DSM-5 has removed amenorrhea, the psychosocial impact and bone health implications merit mention. Discuss how removing amenorrhea influences diagnosis in males and premenarchal females. In addition, cite this publication: https://pubmed.ncbi.nlm.nih.gov/27527115/ Minor Issues 1. Formatting of Statistical Results; Table 1, Table 2; Page 7-8, Rows 295–313 "Mean and ± standard deviations are reported. In bold statistically significant p." Tabular presentation should unambiguously mark missing data, n-values, and significant p-values. Consider supplementing tables with figures for key relationships (e.g., correlation plots). 2. Missing Details for Regression Models; Tables 3, 5; Page 10, Rows 320–340 "The dependent variable is represented by early versus late age at onset." Regression diagnostics (R², residuals analysis, multicollinearity) should be summarized. More transparency and clarity in the presentation of modeling results are needed. 3. Over-reliance on Univariate Comparisons; Methods/Results Sections "Groups were compared by unpaired sample t tests...variables significant at t tests inserted as predictors in a binary logistic regression model." Consider including covariates (e.g., comorbidities, medication use) in multivariate analyses to control for confounding. 4. Limitations, Impact of Medications Not Explicitly Considered; "It is known that these medications can influence biochemical parameters". Medication effects should be analyzed or at least considered as confounders in statistical models, particularly for electrochemical and lipid parameters. 5. Limitations, Supplement Use as Confounder; "...the use of other supplements or substances may have altered some biomarker values...". Consider a more detailed assessment of supplement use, substance use, and their potential effect on biochemical results. Comments on the Quality of English Languagecan be improved
Author Response
First of all we would like to thank the First Reviewer for the precious comments aimed to improve the present manuscript.
Major Issues
- Retrospective Design & Missing Data; Retrospective studies are prone to bias and incomplete data. Significant missing data for certain biochemical markers (e.g., 50 missing values for RBC, MCV, Hb) undermine the statistical power and generalizability. Authors should quantify missing data for each variable and apply imputation methods or sensitivity analyses.
We thank the reviewer for raising this important point. In the revised manuscript, we have quantified the number of missing values for each biochemical marker and performed multiple imputation analyses to account for missing data. These analyses confirmed the results and the reliability of the linear regression models (with Durbin–Watson values consistently within the acceptable range). Similarly, the omnibus tests (<0.05) and Hosmer–Lemeshow tests (>0.5) supported the reliability of the logistic regression models after multiple imputation analyses. However, for the binary logistic regression, Na⁺/K⁺, result was confirmed only in two out of five imputations, suggesting less robustness than linear regression findings. We have reported this aspect in the revised manuscript and acknowledged it as a limitation, while underlining that the findings from linear regression analysis are unchanged.
- Sample Size & Statistical Power; A sample of 76, further subdivided, especially with large numbers of missing data, is underpowered for regression analyses. This limitation should be addressed explicitly, including a post-hoc power calculation. The authors must acknowledge that small sample size and missing data inflate the risk of Type II error and may lead to spurious associations.
In response, we have performed post-hoc power calculations. The statistical power for the cholesterol analyses was only 17% at α = 0.05, confirming that our sample was underpowered for this outcome. Conversely, the Na⁺/K⁺ ratio analysis showed adequate power (90.6%). We have explicitly acknowledged this limitation in the revised Limitations section, emphasizing the increased risk of Type II error and the need for larger samples in future studies.
- Generalizability; the focus on specialized clinics restricts the findings to more severe cases, potentially limiting applicability to community-based or less severe patients. This should be discussed more thoroughly, and future studies should be proposed that include samples from diverse settings.
We have addressed the reviewer’s comment regarding generalizability by explicitly stating in the Limitations section that our sample is drawn from specialized clinics and may represent more severe cases, which could limit applicability to community-based or less severe patients. We also noted that future studies should include more heterogeneous samples from different settings to confirm these findings.
- Definition and Use of Duration of Untreated Illness (DUI); The definition differs across studies; some consider the initiation of any intervention, not necessarily guideline-based. Authors should clarify their definition, compare it to recent literature, and discuss implications. Further, consider the role of subjective versus objective diagnosis dates.
We have clarified the definition of Duration of Untreated Illness (DUI) in the Methods section, using the same approach as in previous studies from our research group, and have acknowledged the existence of alternative definitions in the literature.
- Discussion, Interpretation of Electrolyte Imbalances; The discussion provides plausible but speculative mechanisms (e.g., renin-angiotensin system involvement, supplement misuse) without direct measurement or supporting biochemical markers. Authors should note these are hypotheses and cite relevant literature for each mechanism. Additionally, the clinical significance should be appraised (incident cardiac arrhythmias, mental status changes). and ensure recent publications are considered: https://jamanetwork.com/journals/jamanetworkopen/fullarticle/2798207
We have clarified in the discussion that the proposed mechanisms underlying electrolyte imbalances (e.g., renin-angiotensin system involvement, supplement misuse) are hypotheses, and we have cited relevant literature to support each proposed mechanism. Furthermore, we have added a commentary on the clinical significance, including potential effects on cardiac rhythm and mental status, and we have ensured that recent publications are referenced.
- Discussion, Cholesterol and Cardiovascular Risk; The link between cholesterol and cardiovascular risk in AN is controversial; some reviews caution against extrapolating risk from hypercholesterolemia in AN due to confounders like malnutrition. It is important to clarify the distinction between transient biochemical changes and long-term clinical risk. see also next point (7).
In response to the reviewer’s comment, we have clarified in the discussion that while elevated cholesterol levels were observed in late-onset AN patients, the association with long-term cardiovascular risk is complex and may be influenced by confounding factors such as malnutrition, hormonal alterations, and body composition. We emphasized that transient biochemical changes should not be interpreted as directly associated to clinical risk.
- Limitations, Incomplete Lipid Profile; "...only total cholesterol (TC) was measured, without including a complete lipid profile..." The lack of HDL, LDL, and triglyceride data limits the evaluation of cardiovascular risk and metabolic status. Authors should recommend complete lipid profiling in future studies and acknowledge this as a major limitation. In addition, they should review and cite recent literature that specifically investigate lipid profiles in AN and include this, also in relation to their interpretation. the authors should consider these publications: https://pmc.ncbi.nlm.nih.gov/articles/PMC6842568/ , https://scielo.isciii.es/pdf/nh/v27n3/24_original21.pdf
We have revised the discussion to clarify that, although elevated cholesterol levels were observed in late-onset AN patients, the association with long-term cardiovascular risk is complex and influenced by potential confounders. We explicitly noted the limitation of having measured only total cholesterol and recommended comprehensive lipid profiling in future studies. Additionally, we cited recent literature investigating lipid profiles in AN to support our discussion.
- Introduction, DSM-5 Diagnostic Criteria Adaptation; While DSM-5 has removed amenorrhea, the psychosocial impact and bone health implications merit mention. Discuss how removing amenorrhea influences diagnosis in males and premenarchal females. In addition, cite this publication: https://pubmed.ncbi.nlm.nih.gov/27527115/
We have revised the Introduction to address the psychosocial and bone health implications of removing amenorrhea from DSM-5 criteria and discussed its impact on diagnosis in males and pre-pubertal females, citing the suggested reference.
Minor Issues
1. Formatting of Statistical Results; Table 1, Table 2; Page 7-8, Rows 295–313 "Mean and ± standard deviations are reported. In bold statistically significant p." Tabular presentation should unambiguously mark missing data, n-values, and significant p-values. Consider supplementing tables with figures for key relationships (e.g., correlation plots).
We have addressed the reviewer’s comment by updating the formatting of statistical results in Tables 1 and 2. Missing data, n-values, and statistically significant p-values are now clearly indicated. Additionally, we have included the requested correlation tables in the appropriate section of the supplementary material.
2. Missing Details for Regression Models; Tables 3, 5; Page 10, Rows 320–340 "The dependent variable is represented by early versus late age at onset." Regression diagnostics (R², residuals analysis, multicollinearity) should be summarized. More transparency and clarity in the presentation of modeling results are needed.
We have implemented the requested changes in the text and in Tables 3 and 5. Additionally, we provide the tables summarizing the residual statistics as supplementary material (not included in the main manuscript). The regression diagnostics, including R², residual analysis, and multicollinearity checks, are now clearly summarized to enhance transparency. The absence of multicollinearity is a strength, as it ensures the reliability of our findings despite a relatively small sample.
3. Over-reliance on Univariate Comparisons; Methods/Results Sections "Groups were compared by unpaired sample t tests...variables significant at t tests inserted as predictors in a binary logistic regression model." Consider including covariates (e.g., comorbidities, medication use) in multivariate analyses to control for confounding.
We included medical comorbidities as covariates in the binary logistic regression model. The association between the Na⁺/K⁺ ratio and age at onset resulted again statistically significant (OR = 2.17, p = 0.034), indicating that this finding is reliable even after inserting the presence of comorbid medical conditions as covariate. Additionally, we have included the requested correlation tables in the appropriate section of the supplementary material.
4. Limitations, Impact of Medications Not Explicitly Considered; "It is known that these medications can influence biochemical parameters". Medication effects should be analyzed or at least considered as confounders in statistical models, particularly for electrochemical and lipid parameters.
We have acknowledged in the Results section that only a small number of patients (6 out of 76) received medications potentially affecting metabolism, and due to this limited number, their impact on biochemical parameters was not analyzed. However, we remarked the potential effect of pharmacotherapy in study limitations.
5. Limitations, Supplement Use as Confounder; "...the use of other supplements or substances may have altered some biomarker values...". Consider a more detailed assessment of supplement use, substance use, and their potential effect on biochemical results.
We agree with the reviewer that supplement and substance use could potentially influence biochemical results. However, the use of vitamin B or D supplements within the three months prior to assessment was an exclusion criterion for our study. Lifetime (not current) substance use disorders was reported only in 9 participants. However, we remarked in study limitations that lifetime substance use disorders can affect biochemical results.

Reviewer 2 Report
Comments and Suggestions for Authors
Dear Authors,
Thanks for the study, which aims to evaluate the role of age at onset on the course of Anorexia Nervosa. The study is interesting, but the manuscript needs some revisions:
- Keywords: I would recommend adding what is essential to the study, such as psychiatric comorbidity, cholesterol level, etc., rather than repeating the words in the title of the publication.
- Introduction: I would recommend conducting a broader literature review on the nutritional, health and mental health status of patients with anorexia.
- Materials and methods:
- The optimal approach would be to divide it into subsections.
- The optimal approach would be to divide it into subsections.
- It would be beneficial to provide a more comprehensive description of the clinical variables, including the methodology used to determine psychiatric and medical comorbidities, personality disorders, etc.
- It is recommended that the time of data collection is indicated.
- Results:
- It would be advisable to indicate a reference starting with sodium in Table 1, so that the reader can immediately compare it with the recommended norm.
- Data are missing for 53 patients for sodium, potassium and sodium/potassium ratios. This means that data is only available for 23 patients, who are divided into two groups based on age at onset. This does not allow for any significant conclusions to be drawn.
- Discussion: The discussion should be expanded to include a range of results, as currently only individual ones are being analyzed.
Author Response
First of all we would like to thank the Second Reviewer for the useful comments aimed to improve the present manuscript.
- Keywords: I would recommend adding what is essential to the study, such as psychiatric comorbidity, cholesterol level, etc., rather than repeating the words in the title of the publication.
As keywords, we have decided to include: Anorexia nervosa, clinical variables, psychiatric comorbidity, cholesterol, Na+/K+, childhood/adolescence vs adulthood onset, mortality.
- Introduction: I would recommend conducting a broader literature review on the nutritional, health and mental health status of patients with anorexia.
Thanks for your suggestion. We have revised and expanded the Introduction section of the manuscript to include a broader literature review on the nutritional, physical, and mental health status of patients with anorexia nervosa, as requested.
- Materials and methods:
- The optimal approach would be to divide it into subsections.
We have revised the manuscript according to your suggestions by dividing the Materials and Methods section into subsections, specifically: 2.1. Sample and Study Design, 2.2. Assessment, 2.3. Blood Collection, and 2.4. Statistical Analyses.
- It would be beneficial to provide a more comprehensive description of the clinical variables, including the methodology used to determine psychiatric and medical comorbidities, personality disorders, etc.
We have expanded the description of the clinical variables in the manuscript, providing a more detailed explanation of the methodology used to assess psychiatric and medical comorbidities, personality disorders, and related factors.
- It is recommended that the time of data collection is indicated.
We have included in the Materials and Methods section that a total of 78 subjects were recruited over 18 months (from July 2021 to December 2022) from the Inpatient Clinic of Fondazione IRCCS Ca’ Granda, Ospedale Maggiore Policlinico (Milan) and the Outpatient Clinic for Eating Disorders of Fondazione IRCCS San Gerardo dei Tintori (Monza).
- Results:
- It would be advisable to indicate a reference starting with sodium in Table 1, so that the reader can immediately compare it with the recommended norm.
We have included in the manuscript the reference values for the parameters in Table 1 as requested, starting with sodium, to facilitate immediate comparison with recommended norms.
- Data are missing for 53 patients for sodium, potassium and sodium/potassium ratios. This means that data is only available for 23 patients, who are divided into two groups based on age at onset. This does not allow for any significant conclusions to be drawn.
We have performed post-hoc power calculation. The statistical power for the cholesterol analyses was only 17% with α = 0.05, confirming that our sample was underpowered for this outcome. Conversely, the Na⁺/K⁺ ratio analysis showed adequate power (90.6%). We have explicitly acknowledged this limitation in the revised Limitations section, emphasizing the increased risk of Type II error and the need for larger samples in future studies.
- Discussion: The discussion should be expanded to include a range of results, as currently only individual ones are being analyzed.
We have expanded the Discussion section as requested, integrating a broader range of results, including clinical, metabolic, and psychosocial outcomes, and providing a more comprehensive analysis of the findings.
Round 2
Reviewer 2 Report
Comments and Suggestions for Authors
Dear Authors,
Thank you for the revised and expanded version of the manuscript.